# The Response of Hormones, Reactive Oxygen Species and Nitric Oxide in the Polyethylene-Glycol-Promoted, Salt–Alkali-Stress-Induced Embryo Germination of *Sorbus pohuashanensis*

**DOI:** 10.3390/ijms25105128

**Published:** 2024-05-08

**Authors:** Xiaodong Wang, Hailong Shen, Ling Yang

**Affiliations:** 1State Key Laboratory of Tree Genetics and Breeding, College of Forestry, Northeast Forestry University, Harbin 150040, China; w1764011009xd@163.com; 2State Forestry and Grassland Administration Engineering Technology Research Center of Korean Pine, Harbin 150040, China; shenhl-cf@nefu.edu.cn; 3State Forestry and Grassland Administration Engineering Technology Research Center of Native Tree Species in Northeast China, Harbin 150040, China

**Keywords:** *Sorbus pohuashanensis*, polyethylene glycol, seed priming, embryo germination, salt–alkali stress

## Abstract

Polyethylene glycol can abrogate plant seed dormancy and alleviate salt–alkali stress damage to plants, but its role in embryonic dormancy abrogation and germination in *Sorbus pohuashanensis* is not yet clear. The mechanism by which polyethylene glycol promotes the release of embryonic dormancy may be related to the synthesis and metabolism of endogenous hormones, reactive oxygen species and reactive nitrogen. In this article, germination in indoor culture dishes was used, and the most suitable conditions for treating *S. pohuashanensis* embryos, with polyethylene glycol (PEG) and sodium carbonate (Na_2_CO_3_), were selected. Germination was observed and recorded, and related physiological indicators such as endogenous hormones, reactive oxygen species and reactive nitrogen were measured and analyzed to elucidate the mechanism of polyethylene glycol in alleviating salt–alkali stress in *S. pohuashanensis* embryos. The results showed that soaking seeds in 5% PEG for 5 days is the best condition to promote germination, which can increase the germination rate of embryos under salt–alkali stress by 1–2 times and improve indicators such as germination speed and the germination index. Polyethylene glycol led to an increase in gibberellin (GA), indole-3-acetic acid (IAA), ethylene (ETH), cytokinin (CTK), nitric oxide (NO), soluble protein and soluble sugar in the embryos under salt–alkali stress; increased activities of superoxide dismutase (SOD), peroxidase (POD), catalase (CAT), nitrate reductase (NR) and nitric oxide synthase (NOS) in the embryos; a reduction in the accumulation of abscisic acid (ABA), hydrogen peroxide (H_2_O_2_) and malondialdehyde (MDA). Therefore, it is suggested that the inhibitory effect of polyethylene glycol on the salt–alkali-stress-induced germination of *S. pohuashanensis* embryos is closely related to the response of endogenous hormones, reactive oxygen species and nitric oxide signalling.

## 1. Introduction

Seed initialization, also known as seed priming or a seed conditioning technique, is a widely used strategy that can protect various crops from harmful environmental stresses without significantly affecting their adaptability and productivity [1]. In this technique, polyethylene glycol is the most commonly used initiator. Polyethylene glycol is chemically inert, has no destructive effects on the seed embryo, does not damage proteins and does not penetrate the seed tissue. Treatment with polyethylene glycol can break seed dormancy and promote germination, improve the uniformity of seed growth and, in particular, promote the emergence of seeds with a long dormancy and a hard seed coat. Its effect is particularly significant and can also improve seeds’ ability to withstand stress [2]. Polyethylene glycol regulates the synthesis and degradation of reactive oxygen species and helps plants to cope with salt and alkali stress [3].

Reactive nitrogen (RNS) plays an important regulatory role in the physiological activities of plants, and nitric oxide (NO) is an important gas molecule that regulates seed germination and increases crop yield. It promotes seed germination by regulating abscisic acid (ABA) metabolism and gibberellin (GA) synthesis pathways [4]. During the seed germination of various plant species, NO signalling can influence physiological responses to factors such as salinity [5]. Current research suggests that the NO synthesis pathway in plants operates via nitrate/nitrite-dependent reduction pathways or NO synthase (NOS)-mediated oxidation pathways, with NR-mediated nitrite reduction into NO being the major source of NO [6].

*Sorbus pohuashanensis* is a deciduous small tree species from the Rosaceae family. It is a valuable tree species in northern China, suitable for manufacture production, landscaping and consumption [7]. *S. pohuashanensis* grows in cold-temperate coniferous forests and is a unique accompanying deciduous tree species with a low natural regeneration ability. Its sensitivity to interspecific competition poses a threat to its genetic diversity [8]. Therefore, effective measures should be taken to protect the germplasm resources of *S. pohuashanensis*. NO has been shown to alleviate embryonic dormancy in Sorbus, which is closely related to ethylene biosynthesis (ETH) and ABA degradation metabolism [7]. Polyethylene glycol can abrogate plant seed dormancy and alleviate salt–alkali stress damage to plants, but its role in abrogating embryonic dormancy and germination in *S. pohuashanensis* is not yet clear. It is hypothesized that the mechanism by which polyethylene glycol promotes embryonic dormancy abrogation may be related to the synthesis and metabolism of endogenous hormones, reactive oxygen species and reactive nitrogen.

In this study, polyethylene glycol (PEG) and sodium carbonate (Na_2_CO_3_) were used to treat the embryos of *S. pohuashanensis*. Their germination status was observed and recorded, and related physiological indicators such as endogenous hormones, reactive oxygen species and reactive nitrogen were measured and analyzed to elucidate the mechanism of polyethylene glycol in alleviating saline–alkali stress in *S. pohuashanensis* embryos. The research results provide new insights into the signalling and physiological mechanisms of PEG-induced promotion of embryo germination in *S. pohuashanensis*.

## 2. Results

### 2.1. Effects of PEG and Na_2_CO_3_ on Embryo Germination of S. pohuashanensis

PEG significantly increased the germination percentage, germination speed and germination index of the embryos (*p* < 0.05, Figure 1). Of all the treatments, the 5-day soak with 5% PEG resulted in the best germination of the *S. pohuashanensis* embryos (germination percentage 86.67%), significantly higher than the control and the other treatments. Therefore, soaking the seeds in 5% PEG for 5 days is the condition for subsequent germination in a saline alkali environment.

Saline–alkali stress significantly impairs the embryo germination of *S. pohuashanensis*. The percentage of embryo germination in the Na_2_CO_3_ treatment was only 7.78%, while the germination rates in the PEG and control treatments were 80.00% and 64.44%, respectively, which were significantly higher than those in the saline–alkali stress treatment. The germination rate and germination index of the Na_2_CO_3_-treated embryos were significantly lower than those of the PEG and control treatments (*p* < 0.05, Table 1). The percentage of germination of the embryos in the combined treatment with PEG and Na_2_CO_3_ was significantly higher, by 214.14%, than that in the Na_2_CO_3_ treatment alone, and the other germination indicators were also improved to some extent, but the improvement was not significant compared with PEG and the control (Table 1). Therefore, it is suggested that PEG has some alleviating effect on the germination of *S. pohuashanensis* embryos under salt–alkali stress (Figure 2).

### 2.2. Changes in Endogenous Hormone Content during the Embryo Germination of S. pohuashanensis

With an increasing germination time, the GA content gradually increased in the PEG-treated embryos of *S. pohuashanensis*, while the GA content gradually decreased in the Na_2_CO_3_-treated embryos. From germination to the seedling stage, the GA content was significantly higher in the PEG treatment than in the control and Na_2_CO_3_ treatments (*p* < 0.05, Figure 3A). The GA content of the embryos under PEG-induced Na_2_CO_3_ stress was significantly higher than that of the other treatments (Figure 3A).

The IAA content was significantly higher in the PEG treatment than in the other treatments and reached its maximum at the seedling stage. The IAA content was higher in the early stage of embryo germination in the Na_2_CO_3_ treatment than in the control and lower in the later stage than in the other treatments (Figure 3B). After PEG-induced Na_2_CO_3_ stress, the IAA content of the embryos first increased and then decreased, which was significantly higher than that of the solely Na_2_CO_3_ treatment (Figure 3B).

The ABA content in all the treatments showed a decreasing trend; at the early embryo germination stage, the ABA content in the PEG treatment was significantly lower than in the other treatments; during the seedling germination stage, the PEG treatment was significantly lower than the control and the Na_2_CO_3_ treatment (Figure 3C). The ABA content in the Na_2_CO_3_ treatment remained the highest and was significantly higher than in the other treatments. After the start of the PEG treatment, the ABA content gradually decreased and reached its minimum at the young seedling stage (Figure 3C).

The CTK content in all the treatments showed an upward trend, and the CTK content of the PEG treatment was significantly higher than that of the control and the Na_2_CO_3_ treatment; the CTK content in the Na_2_CO_3_ treatment remained the lowest (Figure 3D). The CTK content in the Na_2_CO_3_ treatment reached its maximum at the seedling stage after PEG initiation and was thus significantly higher than that of the other treatments (Figure 3D). The ETH content in the PEG treatment remained highest and was significantly higher than the other treatments (Figure 3E). After the PEG treatment, the ETH content increased significantly in the Na_2_CO_3_ treatment compared to the Na_2_CO_3_-only treatment (Figure 3E).

In conclusion, PEG induced an increase in the synthesis of growth-promoting hormone in the *S. pohuashanensis* embryos under Na_2_CO_3_ stress and simultaneously reduced the accumulation of growth-inhibiting hormone.

### 2.3. Changes in Reactive Oxygen Species Content during the Embryonic Germination of S. pohuashanensis

During seedling germination, the H_2_O_2_ content in the PEG treatment was significantly lower than that in the control treatment (*p* < 0.05), and the trend was the same every day; the Na_2_CO_3_ treatment significantly triggered the accumulation of H_2_O_2_, which remained the highest among all the treatments; PEG induced a decrease in the accumulation of H_2_O_2_, with its content significantly lower than in the other treatments (Figure 4A). The trend in the MDA content remained consistent among the different treatments in the late stage of germination, and the MDA content in the PEG treatment was significantly lower than that in the other treatments. The MDA content of the Na_2_CO_3_ treatment gradually increased with an increasing germination time and reached its maximum on the eighth day.

### 2.4. Changes in Antioxidant Enzyme Activity during the Embryonic Germination of S. pohuashanensis

With the lifting of embryonic dormancy in *S. pohuashanensis*, the activities of SOD, POD and CAT gradually increased in the seedlings (Figure 5A–C). The SOD activity in the PEG treatment was significantly higher than that in the control treatment (*p* < 0.05), while that in the Na_2_CO_3_ treatment was significantly lower than that in the control treatment. The SOD activity in the combined treatment with PEG and Na_2_CO_3_ was significantly higher than that in the solely Na_2_CO_3_ treatment (Figure 5A). On day 8 of germination, the POD activity in the PEG treatment reached its maximum of 126.67 U/g/min, a significant increase of 46.15% compared to the control. At this time point, the combined PEG and Na_2_CO_3_ treatment increased by 75% compared to the Na_2_CO_3_ treatment, but there was no significant difference compared to the control (Figure 5B). The CAT activity in the Na_2_CO_3_ treatment was significantly lower than that in the control treatment at the late stage of germination, while the CAT activity in the PEG treatment remained the highest. After PEG initiation, the CAT activity in the Na_2_CO_3_ treatment was higher than that in the individual treatments and even exceeded the control level (Figure 5C).

On the last day of seedling germination, there was some increase in the soluble protein and soluble sugar content in the different treatments (Figure 5D,E). The soluble protein content first increased and then decreased in the PEG treatment and the control, while the other treatments showed a gradual increase (Figure 5D). Compared to the control, the PEG treatment significantly increased the soluble protein content in the seedlings. On the 8th day of germination, the soluble protein content in each treatment was graded from high to low: PEG > PEG + Na_2_CO_3_ > Water > Na_2_CO_3_. At this time, the combined PEG and Na_2_CO_3_ treatment was significantly increased by 45.11% compared to the Na_2_CO_3_ treatment (Figure 5D). The trend in the soluble sugar content in the different treatments was consistent with that in soluble protein (Figure 5E). On the 8th day of germination, the lowest soluble sugar content in the Na_2_CO_3_ treatment was 6.82 mg/g, which was not significantly different from the control group. At this time, the combined PEG and Na_2_CO_3_ treatment increased by 81.67% and 62.60% compared to the Na_2_CO_3_ and control treatment, respectively, reaching the level of the PEG treatment (Figure 5E).

### 2.5. Changes in Active Nitrogen Content during the Germination of S. pohuashanensis

In the late phase of germination, the content of active nitrogen in the different treatments showed a general upward trend (Figure 6). The NO content was highest in the PEG treatment and lowest in the Na_2_CO_3_ treatment; on the 8th day of germination, the NO content increased significantly by 31.12% in the PEG treatment compared to the control treatment. The combined PEG and Na_2_CO_3_ treatment significantly increased the NO content by 99.61% and 27.98% compared to the Na_2_CO_3_ treatment and the control treatment, respectively (*p* < 0.05), although the difference was not significant compared to the PEG treatment (Figure 6A).

The NOS activity in the PEG treatment was highest in the early phase of seedling germination, while the NOS activity in the control treatment decreased slightly. At the late stage of seedling germination, the maximum value of the PEG and Na_2_CO_3_ combination treatment was 13.20 U/mg/prot. At this stage, the PEG and Na_2_CO_3_ combination treatment increased significantly by 72.77% and 115.33% compared to the Na_2_CO_3_ treatment and the control treatment, respectively, while there was no significant difference from the PEG treatment (Figure 6B). The NR activity in the Na_2_CO_3_ treatment and the combined PEG and Na_2_CO_3_ treatment showed a gradually increasing trend. The NR activity in the PEG treatment increased by 45.46% compared to the control, and the combined PEG and Na_2_CO_3_ treatment increased by 11.26% compared to the Na_2_CO_3_ treatment (Figure 6C).

## 3. Discussion

The two most important factors in seed priming technology are the initiation time and the degree of re-drying [9]. A saline, alkaline environment not only has a direct effect on seed germination but also limits their growth by affecting their physiological metabolism and photosynthesis [10]. Seed germination is an important indicator of the salt and alkali tolerance of plants [11]. The appropriate concentration of polyethylene glycol has an optimal effect on seed germination under salt–alkali stress [12]. In this study, PEG was found to significantly promote the germination of the embryos of *S. pohuashanensis*, while Na_2_CO_3_ inhibited the germination of the embryos. Appropriate concentrations of PEG can attenuate the inhibition of embryos in a Na_2_CO_3_ environment. However, its positive effect is concentration-dependent. For example, 5% and 15% PEG can promote embryo germination, while 25% and 35% PEG have no significant effect on embryo germination and can even inhibit their growth. Soaking seeds in 5% PEG for 5 days is the best condition to promote the germination of *S. pohuashanensis* embryos. PEG at a low concentration can effectively promote seed germination indicators [13,14]; this is similar to the concentration-dependent effect of exogenous NO on promoting the germination of plant embryos, i.e., low concentrations of NO promote the germination of embryos in Sorbus, while high concentrations inhibit the germination of embryos [15].

The hormone system plays a crucial role in plant development and the ability of plants to adapt to environmental stress [16]. The adaptation of plants to stress is closely linked to the balance of endogenous plant hormones [17]. GA, as an endogenous signalling molecule for breaking seed dormancy, can directly promote the synthesis of endogenous GA, regulate the metabolism of substances and the degradation of storage substances and synergistically regulate seed germination, which is currently one of the most effective methods for regulating seed dormancy [18]. In this study, it was found that under the priming effect of PEG, the GA production of the embryos under stress gradually increased and was even higher than in all the other treatments, while the GA production under the Na_2_CO_3_ treatment gradually decreased to the lowest value, indicating that the GA production gradually decreased with the deepening of salt–alkali stress [19], while PEG priming can prevent this trend. After seed ageing treatment, seven genes encoding IAA were down-regulated. In contrast, PEG treatment promoted gene expression on the BR and IAA signalling pathways, while it inhibited gene expression on the ABA pathway [20]. In this article, the IAA production of the embryos under PEG-induced stress was significantly higher than that under the Na_2_CO_3_ treatment, which is consistent with this conclusion. ABA is an effective seed regulatory hormone that can regulate seed germination and increase the tolerance of different plant species to various stress factors [21]. Studies have shown that ABA has an antagonistic effect on GA, thereby inhibiting seed germination [22]. In this study, it was found that the ABA production of the embryos under PEG-induced stress consistently decreased and was ultimately lower than in all the other treatments, while the ABA production remained highest in the Na_2_CO_3_ treatment, suggesting that PEG can reduce the ABA accumulation caused by abiotic stress. Wang et al. [23] found that under ETH treatment, ACC was continuously produced as germination progressed. However, the ACC production was the lowest in the *S. pohuashanensis* embryos treated with NaHCO_3_, suggesting that salt–alkali stress inhibits the synthesis of the ETH precursor ACC. In this study, the ETH production remained highest in the PEG-treated embryos throughout the germination process, while the ETH production was the lowest in the Na_2_CO_3_-treated embryos. After PEG-induced stress, the ETH production in the embryos increased significantly and reached the control level, which is consistent with previous research results. CTK controls various physiological and biochemical processes in plants and serves as a messenger between and within cells to regulate biotic and abiotic stress [24]. In this study, PEG was found to induce the highest CTK production in the stressed embryos, while the salt–alkali stress treatment resulted in the lowest CTK production, suggesting that PEG induced the production of CTK in the embryos under salt–alkali stress and promoted the growth of the *S. pohuashanensis* embryos.

PEG led to a reduction in reactive oxygen species damage to the embryonic cells. Salt–alkali stress leads to Na+ toxicity and ion imbalance, disrupting important metabolic processes in the plant cells, resulting in the accumulation of reactive oxygen species, which, in turn, leads to protein denaturation, lipid peroxidation, DNA damage, carbohydrate oxidation, pigment degradation and damage to enzyme activity [25]. Under salt–alkali stress, the accumulation of H_2_O_2_ in seeds exacerbates cell membrane damage, leading to poor germination and even seed death [26]. Under salt–alkali stress, the MDA and H_2_O_2_ levels increased significantly in plant seedlings, indicating an exacerbation of oxidative damage [27]. In this study, the PEG treatment was found to reduce the levels of H_2_O_2_ and MDA in the embryos, while the Na_2_CO_3_ treatment significantly increased the levels of H_2_O_2_ and MDA in the embryos compared to the other treatments, resulting in the accumulation of reactive oxygen species. However, PEG induced a significant decrease in the H_2_O_2_ and MDA content in the embryos under salt–alkali stress conditions.

PEG alleviates the inhibitory effect of salt–alkali stress on the embryos of *S. pohuashanensis* by regulating the production of ROS and increasing the activity of antioxidant enzymes and antioxidant levels. Antioxidant enzymes and antioxidants are key factors in the detoxification of reactive oxygen species and maintain the cellular redox balance within physiological limits [28]. To maintain ROS balance and mitigate the damage caused by salt–alkali stress, a reactive oxygen species scavenging system, mainly consisting of antioxidant enzymes, is formed in the plant body [29]. A significant increase in the MDA and H_2_O_2_ levels stimulates the formation of antioxidant defense systems in the plant [30]. Under salt–alkali stress, CAT and SOD play an important role in the defense against oxidative damage and the regulation of the coordinated removal of intracellular ROS by antioxidant enzymes. In this study, the PEG treatment significantly increased the activity of antioxidant enzymes and the antioxidant content in the embryos, thus reducing the accumulation of reactive oxygen species. After the PEG treatment, the decrease in the oxidative stress response of the seedlings under salt–alkali stress is attributed to the increase in intracellular antioxidant enzyme activity. However, research has found that a decrease in POD activity being triggered under salt–alkali stress [31] suggests that the triggering treatment regulates the seedlings’ ability to adapt to salt stress by regulating the activity of some antioxidant enzymes in their bodies, thereby increasing their salt tolerance and protecting the cells from oxidative damage [32].

It has been shown that NO is not only an important source of nitrogen during seed development but is also involved in various plant stress responses to high salinity, drought and high temperatures [5]. Nitric oxide (NO) and nitroso-mercaptan (SNO) are signalling molecules and products of nitrogen metabolism. Nitrate nitrogen (NO_3_^−^) is the main nitrogen source, and nitrate nitrogen transporters (NRTs) are responsible for the uptake or transport of NO_3_^−^. In this study, the NO content increased by 99.61 in the embryos treated with a combination of PEG and Na_2_CO_3_ compared to the Na_2_CO_3_ treatment, suggesting that breaking the dormancy of *S. pohuashanensis* seeds and promoting their germination is the result of the synergistic regulation of hormones such as ABA and NO signalling molecules, supporting the findings of Wang et al. [7]. After treatment with sodium nitroprusside (SNP, NO donor), the salt tolerance index of the plant seedlings increased significantly. Under salt–alkali stress conditions, the NO content in the plant seedlings increased, which enhanced their resistance [33], which is comparable to the results of this study. The NR pathway is the most distinct and important metabolic pathway for NO production [6]. During seed germination, the production of NR-dependent NO is involved in promoting salt and alkali tolerance, suggesting that nitrate promotes plant development and increases stress resistance through the production of NR-related NO [34]. In addition, plants use NOS to catalyse the oxidation of arginine into citrulline and NO [6]. This study found that the NO content and NOS and NR activities were significantly higher in the PEG-treated embryos than in the control group. They were also significantly higher in the combined PEG and Na_2_CO_3_ treatment than in the Na_2_CO_3_ treatment alone. This suggests that nitric oxide plays a positive regulatory role in promoting the germination of *S. pohuashanensis* embryos under polyethylene-glycol-regulated salt–alkali stress.

## 4. Materials and Methods

### 4.1. Experimental Materials

In early October 2022, ripe berries were collected from mature mother trees at the Maoershan Experimental Forest Farm of the Northeast Forestry University (127°30′–127°34′ E, 45°21′–45°25′ N), and the seeds’ quality was assessed using the water selection method. During this process, attention should be paid not to apply too much force so as not to damage the seeds. Plump, pure and ripe seeds with a safe moisture content (9% to 10%) were stored in plastic bags at 0–5 °C for later use.

### 4.2. Experimental Methods

#### 4.2.1. PEG Pretreatment

Before the experiment, dormant seeds with an average mass of 2.44 mg, an average moisture content of 13.83% and an average viability of 93.33% were selected and then soaked in PEG solutions with mass fractions of 0, 5%, 15%, 25% and 35%, respectively, for 1 day, 5 days and 9 days. The following procedure was used: Rinse the swollen seeds with sterile water to remove all the drug residues on the surface of the seeds. Then, use filter paper to soak up the moisture on the surface of the seeds and leave them to dry at room temperature for 48 h. Take seeds that have not been triggered (immersion in sterile water) as a control. After this, the seeds were soaked in distilled water at room temperature and rehydrated for 48 h. After swelling, the seeds were agitated and soaked in a 0.2% (*v*/*v*) NaClO solution for 15 min. The seeds were rinsed with sterile water until they were clear. The seed coat was peeled off on ice, and the naked embryos were squeezed out for the experiment (*S. pohuashanensis* is a seed without an endosperm). The method used to obtain the naked embryos was the same as in the study by Yang Ling et al. [35].

#### 4.2.2. PEG Treatment Germination Test

Place the embryos on double-layered filter paper soaked in 3 mL of sterile water, with sterile water as a control for soaking the seeds. Calculate the time of onset of the germination of the embryos after different treatments, the germination rate for each day during the germination experiment and the cumulative germination percentage for each treatment, as well as the average germination rate, germination index, germination potential and other indicators. Germinate the control and treated embryos in culture dishes with a diameter of 9 cm (30 embryos per dish) under a light intensity of 60 μmol·cm^−2^·s^−1^ at 25 °C (16–8 h of light–darkness per day) under compound white light. Spray the culture tray with sterile water every 24 h until the germination test is completed to maintain humidity. Both the control and the treatment have 3 replicates. Select the best germination condition (5% PEG soak for 5 days) for further analysis.

#### 4.2.3. Salt–Alkali Stress Treatment

In this study, Na_2_CO_3_ (Tianjin Baishi Chemical Co., Ltd., Tianjin, China) was used to simulate the salt–alkali stress environment for the embryos of *S. pohuashanensis*. The experimental concentration was determined based on a soluble salt (Na^+^) content of about 0.15 on the surface of the saline–alkali soil after the experiment (concentration of 14 mmol/L). Prepare the treated seeds with Na_2_CO_3_ and PEG at a mass ratio of 35:1.

#### 4.2.4. Germination Experiment under Salt–Alkali Stress Treatment

Treat the embryos of *S. pohuashanensis* with distilled water, PEG, Na_2_CO_3_ or a mixture of PEG and Na_2_CO_3_. Place the embryos on double-layered filter paper soaked with 3 mL of any solution, observe the indicators, repeat the treatment and germinate them under the same conditions as in Section 4.2.2. The formula for calculating the germination indicators is as follows:Percentage of germination (%) = (a/A) × 100

In the formula, a is the number of embryos germinated during the germination period, and A is the total number of embryos tested.
Average germination rate (d) = Σ (D × X)/Σ X

In the formula, D stands for the number of days calculated from the germination experiment, and X stands for the number of newly germinated embryos on day D. Σ X is the total number of embryos that eventually germinate.
Germination index = Σ Gt/Dt

In the formula, Gt is the daily germination number during the final stage of the germination test; Dt is the number of germination days; Σ is the sum.
Germination capacity (%) = (n/N) × 100

In the formula, n is the number of embryos germinated during the daily peak germination period, and N is the total number of embryos tested.

#### 4.2.5. Endogenous Hormone Determination

The reagent kit uses a one-step dual-antibody sandwich enzyme-linked immunosorbent assay (ELISA) to detect the levels of gibberellin (GA), indole-3-acetic acid (IAA), abscisic acid (ABA), ethylene (ETH) and cytokinin (CTK) in the embryos or seedlings of *S. pohuashanensis*. Add the sample, standard substance and HRP-labelled detection antibodies to the pre-coated wells containing the GA, IAA, ABA, CTH and CTK antibodies, and then incubate and wash them thoroughly. Using the substrate TMB for color development, TMB is converted into blue under the catalysis of peroxidase and finally into yellow under the action of acid. The depth of colour is positively correlated with the GA, IAA, ABA, CTH and CTK in the sample. Measure the absorbance (OD value) using an enzyme-linked immunosorbent assay (ELISA) at a wavelength of 450 nm, and calculate the concentration of each substance based on the logit curve of the ELISA results. Use the natural logarithm of the hormone concentration (ng/mL) as the x-axis and the logit values of the colour values at each concentration as the y-axis. The calculation formula is as follows:Logit (B/B0) = ln (B/B0)/(1 − (B/B0)) = ln B/(B0 − B)

In the formula, B0 stands for the color value of the 0 ng/mL pore and B for other concentrations. Determine the hormone concentration in the sample (ng/mL) based on the logit value of its color rendering value.

#### 4.2.6. Determination of Reactive Oxygen Species Content

The malondialdehyde (MDA) content in the seedlings of *S. pohuashanensis* was determined using the thiobarbituric acid method [36]. The hydrogen peroxide (H_2_O_2_) content was determined according to the method of Peng Yan et al. [37].

#### 4.2.7. Determination of Antioxidant Enzyme Activity

The activity of superoxide dismutase (SOD) in the seedlings of *S. pohuashanensis* was determined using the nitrogen blue tetrazole photoreduction method, the activity of peroxidase (POD) using the guaiacol colorimetric method and the activity of catalase (CAT) using UV spectrophotometry [38]. The soluble protein content was determined using the Coomassie Brilliant Blue G250 method and the soluble sugar content using the Anthron colourimetric method [36].

#### 4.2.8. Determination of the Active Nitrogen Content

The content of nitric oxide (NO) and the activity of nitrate reductase (NR) in the seedlings of *S. pohuashanensis* were determined using a special reagent kit from Suzhou Keming Biotechnology Co., Ltd., Shanghai, China. The activity of nitric oxide synthase (NOS) was measured using a reagent kit from Nanjing Urban Construction Biotechnology Research Institute.

### 4.3. Data Analysis

The data preparation and statistical analysis were performed using Excel 2019 and SPSS 26.0, respectively. SigmaPlot 14.0 was used for the graphical representation. For the analysis of variance, the test of significance was performed at the level of *p* = 0.05, and Duncan’s multiple comparison method was used to test for statistical differences between the means.

## 5. Conclusions

An appropriate concentration of PEG promotes the release and germination of dormancy in the embryos of *S. pohuashanensis*. The saline–alkaline environment created by Na_2_CO_3_ inhibits the germination of the *S. pohuashanensis* embryos, while PEG reduces this inhibitory effect. The alleviating effect of PEG regulates the germination of *S. pohuashanensis* embryos in a saline–alkaline environment mainly by regulating the balance of endogenous hormones, reducing the accumulation of ROS and enhancing antioxidant enzyme activity and NO synthesis and metabolism. The polyethylene glycol priming method may be considered for future application in the cultivation of *S. pohuashanensis* seedlings in saline–alkali soil to alleviate the pressure of salt–alkali stress, promote embryo germination, improve seedling uniformity and increase the seedling activity, antioxidant metabolism level, and stress resistance, thereby reducing the oxidative damage of *S. pohuashanensis* seedlings under salt–alkali stress. The results of this study provide clues for the cultivation of stress-resistant tree species and the protection of their germplasm resources.

## Figures and Tables

**Figure 1 ijms-25-05128-f001:**
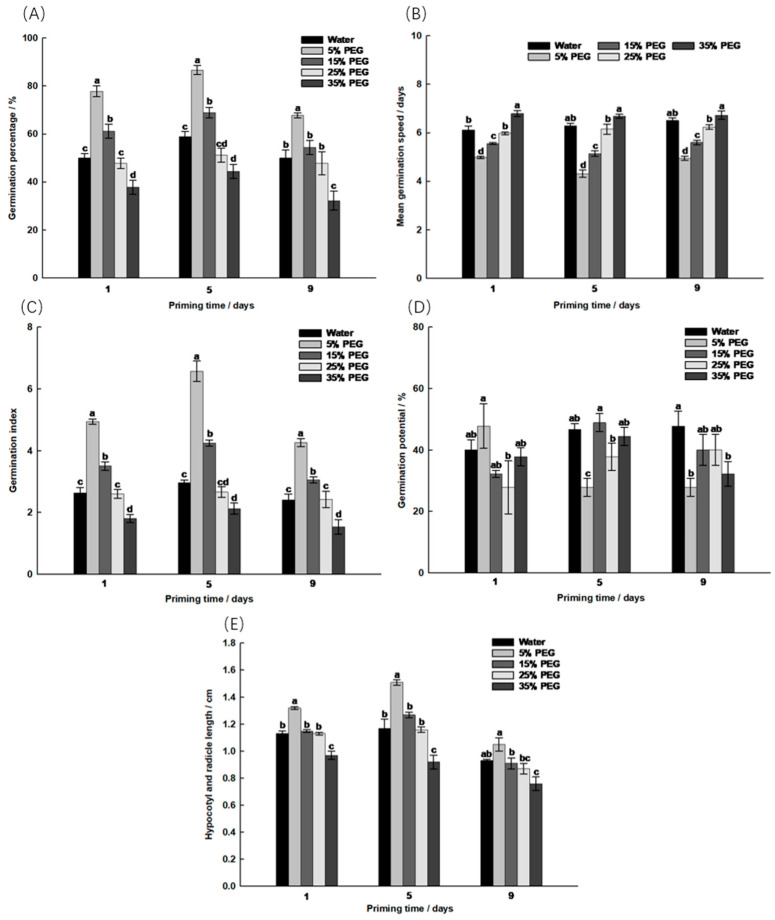
Effect of different polyethylene glycol treatments on the germination of *Sorbus pohuashanensis* embryos. (**A**) Germination percentage; (**B**) Mean germination speed; (**C**) Germination index; (**D**) Germination potential; (**E**) Hypocotyl and radicle length. The mean values followed by different lowercase letters are significantly different according to Duncan’s *t*-test (*p* < 0.05) (mean ± SD; *n* = 3).

**Figure 2 ijms-25-05128-f002:**
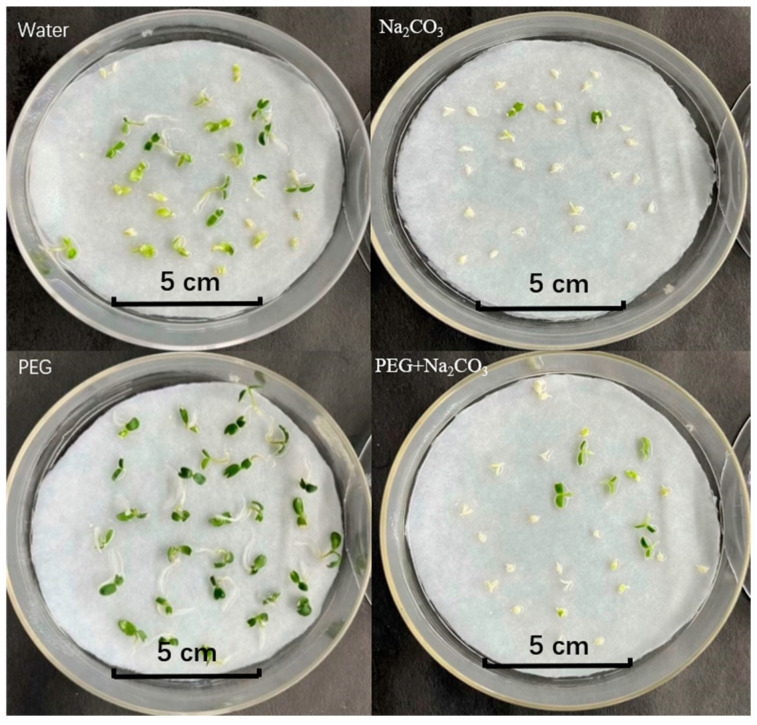
Growth of the seedlings of *Sorbus pohuashanensis* on day 8 of the different treatments. (Bar = 1.0 mm).

**Figure 3 ijms-25-05128-f003:**
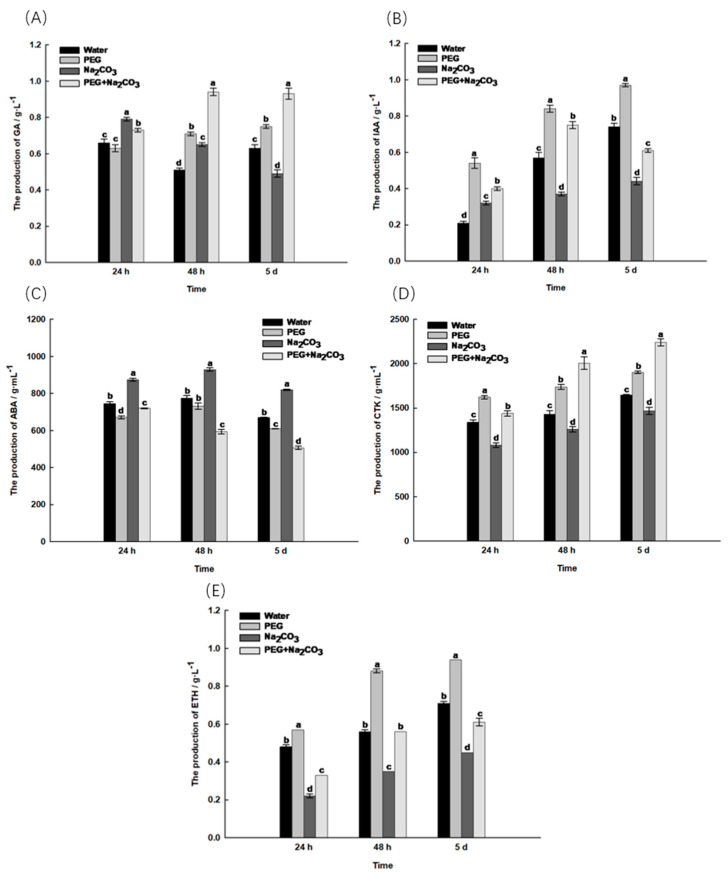
Changes in the endogenous hormone content of *Sorbus pohuashanensis* embryos during germination. (**A**) Gibberellin (GA); (**B**) Indole-3-acetic acid (IAA); (**C**) Abscisic acid (ABA); (**D**) Cytkinin (CTK); (**E**) Ethylene (ETH). The mean values followed by different lowercase letters are sinificantly different according to Duncan’s *t*-test (*p* < 0.05) (mean ± SD; *n* = 3).

**Figure 4 ijms-25-05128-f004:**
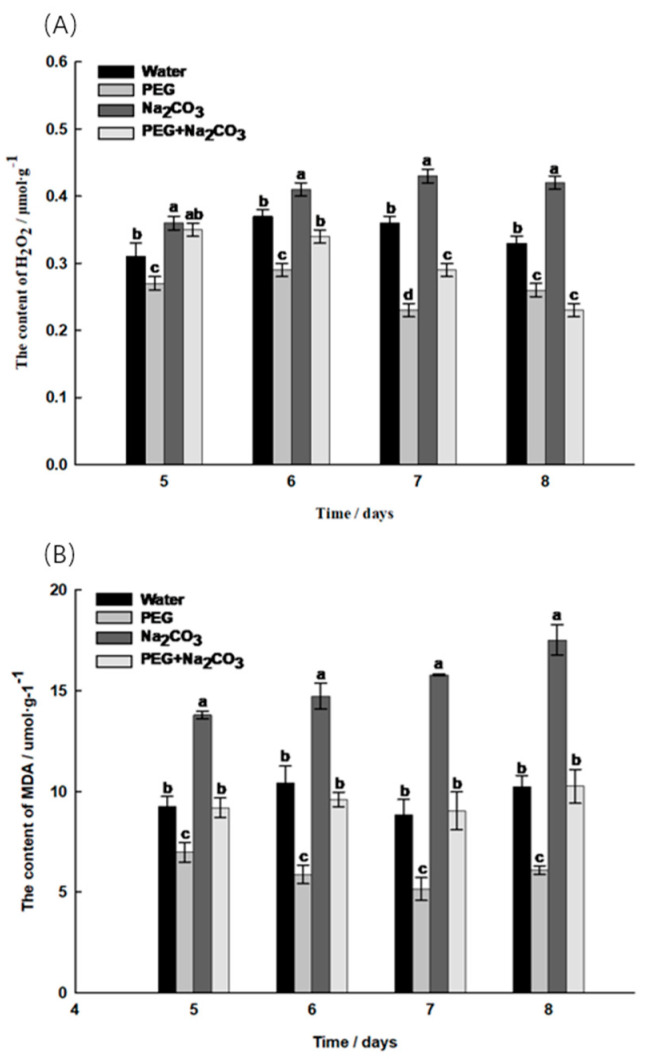
Determination of reactive oxygen species indicators on day 5 to 8 of germination of *Sobus pohuashanensis* seedlings. (**A**) Hydrogen peroxide (H_2_O_2_); (**B**) Malondialdehyde (MDA). The mean values followed by different lowercase letters are significantly different according to Ducan’s *t*-test (*p* < 0.05) (mean ± SD; *n* = 3).

**Figure 5 ijms-25-05128-f005:**
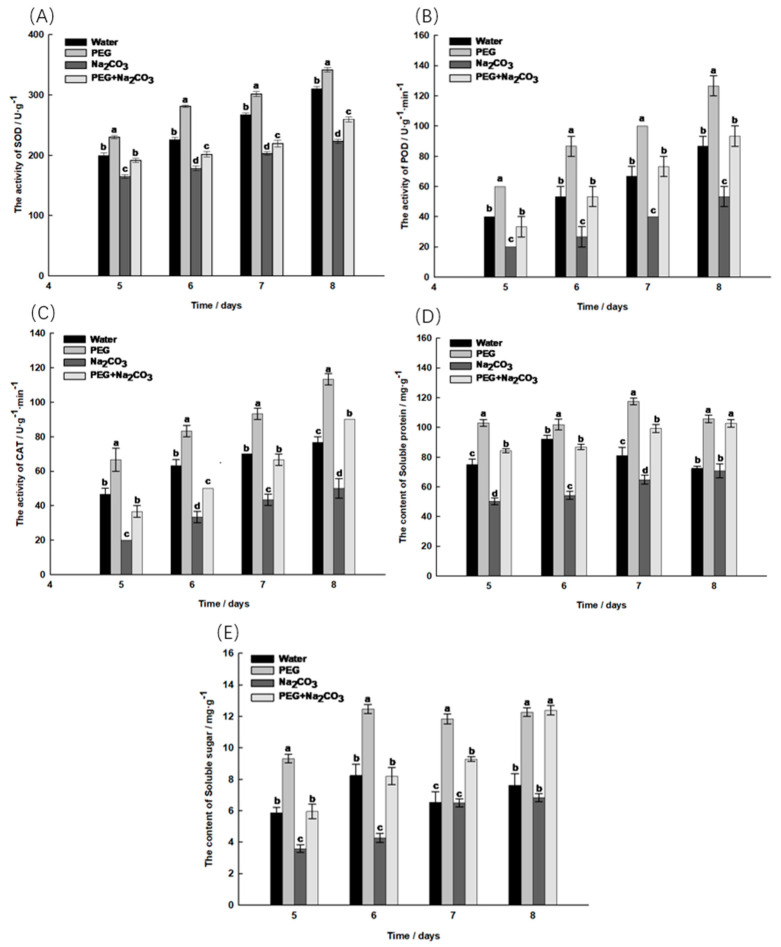
Determination of antioxidant enzyme activity and related indicators on day 5 to 8 of gemination of *Sorbus pohuashanensis* seedlings. (**A**) Superoxide dismutase (SOD); (**B**) Peroxidase (POD); (**C**) Catalase (CAT); (**D**) Soluble protein; (**E**) Soluble sugar. The mean values followed by different lowercase letters are significantly different according to Duncan’s *t*-test (*p* < 0.05) (mean ± SD; *n* = 3).

**Figure 6 ijms-25-05128-f006:**
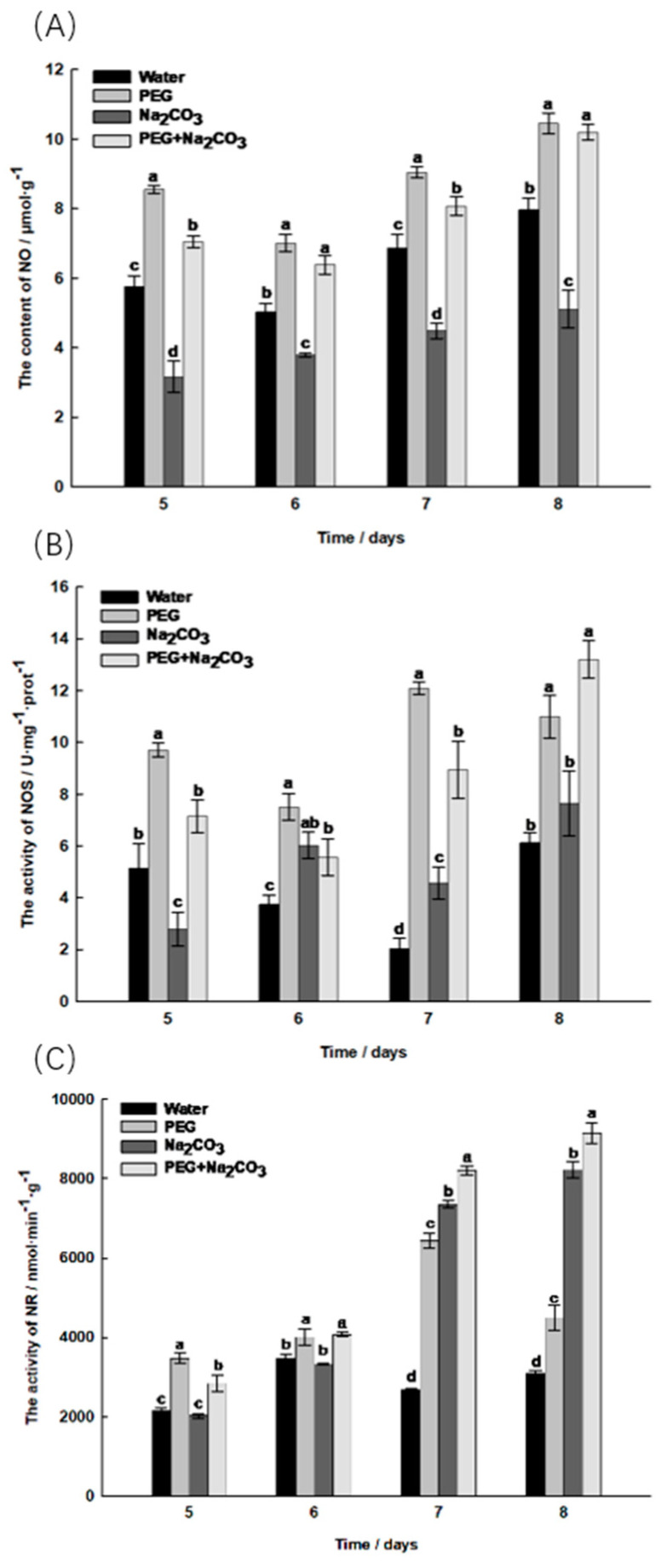
Determination of active nitrogen-related indicators on the 5th to 8th day of germination of *Sorbus pohuashanensis* seedlings. (**A**) Nitric oxide (NO); (**B**) Nitric oxide synthase (NOS); (**C**) Nitrate reductase (NR). The mean values followed by different lowercase letters are significantly different according to Duncan’s *t*-test (*p* < 0.05) (mean ± SD; *n* = 3).

**Table 1 ijms-25-05128-t001:** Effect of salt–alkali stress and polyethylene glycol treatment on the germination indices of *Sorbus pohuashanensis* embryos. Here, 5% PEG soaking for 5 days is the optimum condition. The mean values followed by different lowercase letters are significantly different according to Duncan’s *t*-test (*p* < 0.05) (mean ± SD; *n* = 3).

Index	Water	PEG	Na_2_CO_3_	PEG + Na_2_CO_3_
Germination percentage/%	64.44 ± 5.56 b	80.00 ± 6.94 a	7.78 ± 2.94 d	24.44 ± 1.11 c
Mean germination speed/days	5.64 ± 0.08 b	4.55 ± 0.05 c	6.58 ± 0.36 a	6.06 ± 0.38 ab
Germination index	3.64 ± 0.26 b	5.44 ± 0.44 a	0.35 ± 0.11 d	1.25 ± 0.04 c
Germination potential/%	34.44 ± 2.94 b	65.56 ± 14.57 a	6.67 ± 3.33 c	20.00 ± 3.85 bc
Hypocotyl and radicle length/cm	1.13 ± 0.07 a	1.23 ± 0.25 a	0.38 ± 0.02 b	0.39 ± 0.04 b

## Data Availability

Data contained within the article.

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
