# Peer review of "The Response of Hormones, Reactive Oxygen Species and Nitric Oxide in the Polyethylene-Glycol-Promoted, Salt–Alkali-Stress-Induced Embryo Germination of Sorbus pohuashanensis"

_ijms, 2024, doi:10.3390/ijms25105128_

Round 1

Reviewer 1 Report

Comments and Suggestions for Authors

The authors submitted an article to IJMS concerning the use of PEG in regulating/controlling embryonic dormancy and germination of the plant Sorbus pohuashanensis even under salt and alkali stress, the cultivation of which is important both in maintaining the landscape and from an economic point of view given its consumption. This interesting work, which fits well into the field of research on how to improve the germination of plants that naturally have a low regeneration ability even under difficult conditions such as those that can occur as a result of climate change. The manuscript is well edited, in every part, right from the abstract. The introduction is adequate and refers to recent scientific literature in the field. The results are well described, so that they are clear to the reader, who is also aided in the comprehension of these by well readable graphs. the only note in this section is that in my opinion it should also be made clear in this paragraph that the best conditions as concentration and treatment time with PEG are those 5 % PEG for 5 days and that these are the ones used for subsequent investigations. In this respect at line 80 % is lost.

I appreciate the choice of the many biological variables, right for this study. The discussion is very thorough and consistent with the results, the conclusions are appropriate. 

A note to the authors, in table 1 for the calcium carbonate molecule the numbers should be subscripted.

Author Response

Dear Reviewer,

We are very grateful to you for taking the time to read and modify our article again. We find that your comments play a very important role in improving the quality of our papers. We have carefully revised the paper in light of your comments, and please find our response to the comments made below. We marked the modified part of the manuscript in red.

Thank you for considering our revised manuscript!

Point 1: The only note in this section is that in my opinion it should also be made clear in this paragraph that the best conditions as concentration and treatment time with PEG are those 5 % PEG for 5 days and that these are the ones used for subsequent investigations. In this respect at line 80 % is lost.

Response 1: Thank you very much for your suggestion. We read the manuscript carefully and accepted your suggestions. For details please see the line 82-84.

Point 2: A note to the authors, in table 1 for the calcium carbonate molecule the numbers should be subscripted.

Response 2: Thank you very much for your suggestion. We read the manuscript carefully and accepted your suggestions. For details please see the Table 1.

Reviewer 2 Report

Comments and Suggestions for Authors

Dear Authors,

The experiments were properly planned and conducted in accordance with generally accepted research methodology. The results obtained were analyzed statistically and correctly interpreted. The conclusions are consistent with the arguments presented. The research results are interesting and make a new contribution to the development of science. Nevertheless, I have a few comments:

- The research hypothesis and purpose of the study are missing - please complete.

- Please provide the type of experimental design in which the experiment was conducted, as ANOVA is calculated differently for each experimental design.

- The figures need improvement, as they are illegible.

- The notation of units of measurement should be in accordance with the SI system

- In the title of Table 1, the Authors provide "different lower case letters indicate significant differences at the P = 0.05 level.", and under the table there is " "the mean values followed by different lowercase letters are significantly different according to the Duncan t-test (P < 0.05) (mean ± SD; n = 3)." This is a repetition and should be corrected.

- The information under the figures "letters show significant differences at P =0.05" is not precise enough. In my opinion, it is more appropriate to write "the mean values followed by different lowercase letters are significantly different according to the Duncan t-test (P < 0.05) (mean ± SD; n = 3)", as in L 103-104.

- L 359: „3 or more replicates”? In the description of the figures and table, the authors provide n=3.

Author Response

Dear Reviewer,

We are very grateful to you for taking the time to read and modify our article again. We find that your comments play a very important role in improving the quality of our papers. We have carefully revised the paper in light of your comments, and please find our response to the comments made below. We marked the modified part of the manuscript in yellow highlight.

Thank you for considering our revised manuscript!

Point 1: The research hypothesis and purpose of the study are missing - please complete.

Response 1: Thank you very much for your suggestion. We read the manuscript carefully and accepted your suggestions. Research hypotheses can be found on lines 13-17; Please refer to lines 70-76 for research purposes.

Point 2: Please provide the type of experimental design in which the experiment was conducted, as ANOVA is calculated differently for each experimental design.

Response 2: Thank you very much for your suggestion. We read the manuscript carefully and accepted your suggestions. This experiment adopts one-way analysis of variance and belongs to the completely randomized design type.

Point 3: In the title of Table 1, the Authors provide "different lower case letters indicate significant differences at the P = 0.05 level.", and under the table there is " "the mean values followed by different lowercase letters are significantly different according to the Duncan t-test (P < 0.05) (mean ± SD; n = 3)." This is a repetition and should be corrected.

Response 3: Thank you very much for your suggestion. We accepted your suggestions. For details please see the Table 1.

Point 4: The information under the figures "letters show significant differences at P =0.05" is not precise enough. In my opinion, it is more appropriate to write "the mean values followed by different lowercase letters are significantly different according to the Duncan t-test (P < 0.05) (mean ± SD; n = 3)", as in L 103-104.

Response 4: Thank you very much for your suggestion. We read the manuscript carefully and accepted your suggestions. Please refer to each picture question for details.

Point 5: L 359: “3 or more replicates”? In the description of the figures and table, the authors provide n=3.

Response 5: Thank you very much for your suggestion. We read the manuscript carefully and accepted your suggestions. For details please see the line 363-364. This experiment was repeated 3 replicates, which may have been due to my incorrect expression and caused your misunderstanding. I apologize for any inconvenience caused.

Point 6: L The figures need improvement, as they are illegible.

Response 6: Thank you very much for your suggestion. We read the manuscript carefully and accepted your suggestions. The clarity of the figures has been modified. Please refer to it.

Point 7: L The notation of units of measurement should be in accordance with the SI system.

Response 7: Thank you very much for your suggestion. We read the manuscript carefully and accepted your suggestions. The notation of the measurement unit in this article conforms to the SI system.